

# Microsatellite loci discovery from next-generation sequencing data and loci characterization in the epizoic barnacle *Chelonibia testudinaria* (Linnaeus, 1758)

Christine Ewers-Saucedo[1,2], John D. Zardus[3] and John P. Wares[2,4]

[1] Evolution and Ecology, University of California, Davis, CA, United States
[2] Department of Genetics, University of Georgia, Athens, GA, United States
[3] Department of Biology, The Citadel, Charleston, SC, United States
[4] Odum School of Ecology, University of Georgia, Athens, Georgia, United States

## ABSTRACT

Microsatellite markers remain an important tool for ecological and evolutionary research, but are unavailable for many non-model organisms. One such organism with rare ecological and evolutionary features is the epizoic barnacle *Chelonibia testudinaria* (*Linnaeus, 1758*). *Chelonibia testudinaria* appears to be a host generalist, and has an unusual sexual system, androdioecy. Genetic studies on host specificity and mating behavior are impeded by the lack of fine-scale, highly variable markers, such as microsatellite markers. In the present study, we discovered thousands of new microsatellite loci from next-generation sequencing data, and characterized 12 loci thoroughly. We conclude that 11 of these loci will be useful markers in future ecological and evolutionary studies on *C. testudinaria*.

## INTRODUCTION

Microsatellite loci are valuable tools in ecological and evolutionary studies (e.g. *Jarne & Lagoda, 1996*; *Vignal et al., 2002*; *Selkoe & Toonen, 2006*). Next-generation sequencing approaches have revolutionized microsatellite loci development, allowing the rapid discovery of thousands of potential microsatellite loci in the genome of non-model organisms (*Castoe et al., 2012*). However, a thorough characterization of potential microsatellite loci remains labor-intensive. It is nonetheless necessary to do so if we want to use these markers successfully in evolutionary and ecological studies.

A non-model organism for which genetic and genomic resources are lacking is the epizoic barnacle *Chelonibia testudinaria* (*Linnaeus, 1758*). *Chelonibia testudinaria* uses diverse marine animals as substratum, such as sea turtles, manatees, swimming crabs, and horseshoe crabs. Host-specific morphotypes were previously described as distinct species (*Darwin, 1854*; *Hayashi, 2013*). Recent molecular analyses indicate that *C. testudinaria* is a host generalist, and *C. patula* (*Ranzani, 1818*) and *C. manati* (*Gruvel, 1903*) are now considered synonyms of *C. testudinaria* (*Cheang et al., 2013*; *Zardus et al., 2014*).

Corresponding author
Christine Ewers-Saucedo,
ewers.christine@gmail.com

However, a fine-scale genetic assessment of host-specificity based on highly polymorphic nuclear markers, such as microsatellite markers, is still lacking.

While host species do not seem to provide barriers to gene flow, biogeography does: Three major genetic lineages are restricted to the Indo-West Pacific, Tropical Eastern Pacific and Atlantic Ocean, respectively (*Rawson et al., 2003*; *Cheang et al., 2013*; *Zardus et al., 2014*). These lineages likely represent separate species based on their levels of genetic differentiation (*Zardus et al., 2014*).

All lineages of *C. testudinaria* exhibit a rare sexual system: androdioecy. Androdioecy is characterized by the co-existence of hermaphrodites and males in the same reproductive population. Understanding mating success and mating patterns of both sexes would greatly advance our understanding of this rare sexual system. This would be most easily achieved with genetic parentage assignment–but its prerequisite, highly variable genetic markers, are not available.

In order to overcome these shortcomings, we used next-generation sequencing to discover microsatellite markers for *C. testudinaria*. We characterized 12 promising markers in populations from the Atlantic coast of the United States and from the Northeastern coast of Australia.

## MATERIAL AND METHODS

### Specimen collections

Specimen collections of the Atlantic lineage took place on Nannygoat Beach, Sapelo Island, GA, USA (31.48 °N, 81.24 °W) between 2012 and 2014 under the collection permit of the University of Georgia Marine Institute, and sanctioned by the Georgia DNR Wildlife Services. We chose to collect from the horseshoe crab *Limulus polyphemus* (*Linnaeus, 1758*) because it is relatively abundant and easy to sample: each spring and early summer, horseshoe crabs crawl onto beaches to mate and lay their eggs. During this process, we removed one individual of *C. testudinaria* per host individual with a sharp knife directly on the beach, and preserved it in 95% EtOH immediately after collection. We collected specimens of the Indo-West Pacific lineage in the vicinity of Townsville, Queensland, Australia (23 °S, 143 °E), in September 2012 from green turtles (*Chelonia mydas* Brongniart, 1800). Working with officials from the Queensland Department of Environment and Heritage Protection, turtles were captured in-water for routine tagging and release. During capture barnacles were collected and immediately preserved in 95% EtOH.

### Microsatellite loci discovery

We extracted genomic DNA from the feeding appendage of a single large hermaphroditic *C. testudinaria* collected from a horseshoe crab with Gentra Puregene Tissue Kit (Qiagen), and measured DNA concentration with a Qubit 2.0 Fluorometer (Life Technologies, Carlsbad, CA, USA). Genomic DNA was fragmented into approximately 700 bp lengths (insert size) and shotgun-sequenced on an Illumina MiSeq sequencer (PE250). We quality-checked paired-end reads with FastQC (*Andrews, 2015*). The software FASTQMCF was used to trim adapters, cut low quality ends and remove low quality reads and their mate-pair read (*Aronesty, 2011*).

We executed the perl script PALFINDER to identify short sequence repeat regions (*Castoe et al., 2012*). The script also calls PRIMER3, version 2.0.0, to identify potential primer pairs that span the repeat region (*Rozen & Skaletsky, 2000*). The minimum number of repeat units was chosen as in *Castoe et al. (2012)*. A repeat unit, also called kmer, is defined as the length of the short sequence repeat. For example, a dimer would be a repeat of two base pairs (e.g. GC), and a tetramer would be a repeat of four basepairs (e.g. AGGT). PRIMER3 parameters were the default values. The search resulted in a large number of potentially amplifiable loci (PALs), repeat regions for which primers were identified. We filtered the results by removing all PALs which occurred less than two times and more than the estimated genome coverage in the genomic reads based on the following reasoning: If the number of primer occurrences is low, the primer sequence may contain sequencing error. If the number of occurrences is higher than the expected genome coverage, the primer region may occur more than once in the genome, leading to amplification of multiple loci (genomic regions). Neither of these outcomes is desirable because a good marker occurs only once in the genome, and has a primer sequence that matches the genomic sequence well. We estimated genome coverage by mapping our genomic reads to 52 nuclear single-copy gene fragments available from the acorn barnacle *Semibalanus balanoides* (*Regier et al., 2010*) with the Geneious version 6.0.3 Read Mapper, using the default settings (*Kearse et al., 2012*), calculated the median coverage for each of the 52 gene fragments, and the grand median of all genes. R scripts for screening PALFINDER output as well as calculating genome coverage are available as Supplementary Information.

Of the filtered PALs, we chose 48 PALs for trial amplification, which differed in kmer length, kmer motif (e.g. AG vs TG) and fragment size. We extracted and amplified DNA of 16 *C. testudinaria* individuals for trials. DNA was extracted from feeding appendages of barnacles with the Chelex method (*Walsh, Metzger & Higuchi, 1991*). Trials used the method of *Schuelke (2000)* to amplify fragments and simultaneously tag forward primers with a fluorescent dye. Loci that amplified and scored consistently in all individuals were fluorescently labeled with 6-FAM, NED or HEX (Applied Biosystems, Custom Oligo Synthesis Center), and used on a larger number of individuals to characterize the microsatellite loci.

## Microsatellite loci amplification

Genomic DNA was extracted from feeding appendages of barnacles with the Chelex method (*Walsh, Metzger & Higuchi, 1991*). PCR amplifications were performed in 20 ul volumes containing final concentrations of $1\times$ PCR buffer (Bioline), 5% bovine serum albumin 10 mg/mL (Sigma), 200 mM each dNTP, 2 nM MgCl, 0.5 mM each primer, 0.5 units of Promega GoTaq DNA Polymerase, and 1 μl template DNA. PCR conditions were as follows: 4 min initial denaturation, followed by 40 cycles with 45 sec denaturing at 94 °C, 60 sec annealing at 55 °C, 60 sec extension at 72 °C and a final extension time of 10 min. The PCR were carried out in a MJ Research PCR Engine. HiDi and ROX500 size standard were added to each sample, and fragment length analysis was carried out at the Georgia Genomics Facility on an ABI 3730xl. Peaks were called and binned with the microsatellite plugin of Geneious version 8.1 (*Kearse et al., 2012*).

## Microsatellite loci characterization

We first characterized the microsatellite loci on 42 individuals from the Atlantic population. We inspected peak calls for fragment size consistency, using the R package MSATALLELE (*Alberto, 2009*). MSATALLELE plots peak calls of a locus in histogram form, facilitating visual binning of alleles. If bins could not be clearly assigned, the locus was excluded from the subsequent analysis. We tested whether loci were in Hardy-Weinberg Equilibrium (HWE) by using 1999 Monte Carlo permutations, as implemented by the function HW.TEST in the R package PEGAS (*Paradis, 2010*). Significance values were adjusted for multiple comparisons based on the method of *Holm (1979)*. We recorded the number of alleles, range of fragment sizes, and allelic richness of each locus. The frequency of null alleles was computed based on the method of *Brookfield (1996)*. Genotyping error rates were calculated by repeating genotyping for all individuals. After characterizing the loci in the Atlantic population, we amplified the loci for 24 individuals from Queensland, Australia, to assess if the loci could be used in cross-lineage analysis. Characterization on individuals of the Indo-West Pacific population were largely the same as for the Atlantic population, but we did not assess genotyping error rates. A R script detailing these analyses is available as Supplementary Information.

# RESULTS

## Microsatellite marker development

The MiSeq run generated 15,324,079 paired-end reads (35–251 bp long) with 81.05% > Q30. Raw reads are available in NCBI's short read archive (Study accession: PRJNA310774, run accession: SRR3144544). After quality control, 13,498,280 paired-end reads (19–251 bp long) remained, for a total of 6.2 Gb. The median genome coverage was $8\times$ (min = 3, max = 24) for 52 nuclear single-copy gene fragments. The PALFINDER script detected 629,990 microsatellite repeat regions, of which 29,627 (5.38%) were potentially amplifiable loci (PALs) with forward and reverse primer. A summary of detected microsatellite repeat regions is available as Supplementary Information. A list of all detected microsatellite regions (with primer sequences) is available on figshare (DOI: 10.6084/m9.figshare.2070070). After removing PALs with more than eight or less than two occurrences of either forward or reverse primer in the sequence read data, 17,265 PALs remained. We chose 48 perfect repeat loci for trial amplification, which differed in kmer length and repeat motif, but were otherwise chosen at random. Of those 48 loci, 12 loci amplified and scored consistently throughout the trials, and were tagged with fluorescently labeled dye (Table 1).

## Microsatellite marker characterization

We genotyped 42 individuals successfully at more than half of the 12 consistently scoring loci. Visual inspection of peak call histograms revealed that peak calls of Ctest2 did not have clearly defined bins, and were excluded from subsequent analyses. The number of alleles of the 11 scorable loci ranged from six to thirty (Fig. 1). Microsatellite genotype and collecting date for each individual are available as Supplementary Information. For the Atlantic population, four loci were not in HWE, and showed homozygote

**Table 1 Microsatellite loci amplification information.** All loci were amplified at 55 °C annealing temperature. "Dye" refers to the fluorescent color label for each forward primer. NED is yellow, 6-FAM is blue and HEX is green. Labeling forward primers with different colors allows multiplexing several primer sets in the same reaction. "Multiplex reaction" refers to the multiplexing PCR scheme, e.g. all loci with the same multiplex code were amplified in the same reaction.

| Locus | Kmer | Motif | Forward primer sequence | Reverse primer sequence | Dye | Multiplex reaction |
|-------|------|-------|-------------------------|-------------------------|-----|--------------------|
| Ctest2 | | TGC | ACACACATCACTGGACTCG | CAGTAAGCAGCTCTGTTCG | NED | BB |
| Ctest7 | 4 | TCCG | GTTATCCGTCATTCCATCC | GACGTAACCACCTTGTCG | 6-FAM | AA |
| Ctest9 | 4 | AATC | AACAGATGTGACATTGATGC | TTGTACTGTCCTTGTAACGC | 6-FAM | BB |
| Ctest10 | 2 | AC | ATACGCACAAACTCACACC | TGTCCTCTTACAGAGATCGG | HEX | BB |
| Ctest11 | 2 | TG | GTGTCCACCTTTATGTCTGG | AGTTGAAAATACGCACGC | HEX | CC |
| Ctest12 | 4 | TCCG | AACTGGTGGACAGTCTGG | CATCTTTATGAGTAGCGAGG | HEX | AA |
| Ctest16 | 4 | AGCC | TCAGGTACAGCATTATCGC | CAAGGACCATCAATTACCC | 6-FAM | CC |
| Ctest18 | 5 | AGGTC | TTCATGAATCACTTCCTGG | GTAATCAAATAAGGCGATGC | NED | AA |
| Ctest31 | 4 | AACT | GTACGCCGAAAGTAAAGC | AGCTCTGACAAAGTTATGCC | 6-FAM | DD |
| Ctest32 | 4 | TCTG | AGAAATCCATAATCGTCTGG | ATAACGACGTAATCAGCACC | NED | CC |
| Ctest36 | 4 | ATAC | AGATATTGGTGGAACGAGC | CACAACATACTCAACGAACG | HEX | DD |
| Ctest47 | 5 | TCGTG | GTTGACACGATGACATAACG | ACAATTCCAGCTCTGTTAGC | NED | DD |

excess (Table 2). The estimated percentage of null alleles ranged from 0–23%. Allelic richness ranged from 3.65–20.7, and genotyping error rates ranged from 0–7% (Table 2).

All 11 loci amplified in some of the 23 Australian individuals, and had at least two alleles (Figure 2). No loci amplified in all individuals, and no individual failed to amplify all loci. For the Pacific lineage of *C. testudinaria*, two loci deviated significantly from HWE (Table 3). The locus Ctest7 showed heterozygote excess, even though not at a significant level (p-value = 0.13). Allelic richness of the Australian population was overall lower than in the Atlantic population. The percentage of null alleles ranged from 0–33%.

## DISCUSSION

The present study developed and characterized 11 microsatellite markers for the epizoic barnacle *Chelonibia testudinaria*. Several loci are not in HWE, probably due to null alleles. Their high allelic diversity and scoring consistency should nonetheless make them useful in ecological and evolutionary studies. In addition, we provide the resources to evaluate thousands of additional potentially amplifiable loci (PALs) for *C. testudinaria*.

Several loci were not in HWE, and displayed homozygote excess. Homozygote excess can have several causes: selection on these loci, the presence of null alleles, inbreeding, population substructure or large variance in reproductive success. Inbreeding is unlikely because most barnacles are obligate outcrossers and *C. testudinaria* has a widely-dispersing planktonic larval phase. Selection cannot be excluded as an explanation, but selection on or near several markers appears unlikely. Population substructure may be present, but if so, is neither host-induced nor geographical. Large variance in reproductive success can cause homozygote excess (*Hedgecock, 1994*), and has been invoked to explain homozygote excess in e.g. sea urchins (*Addison & Hart, 2004*). If variance in

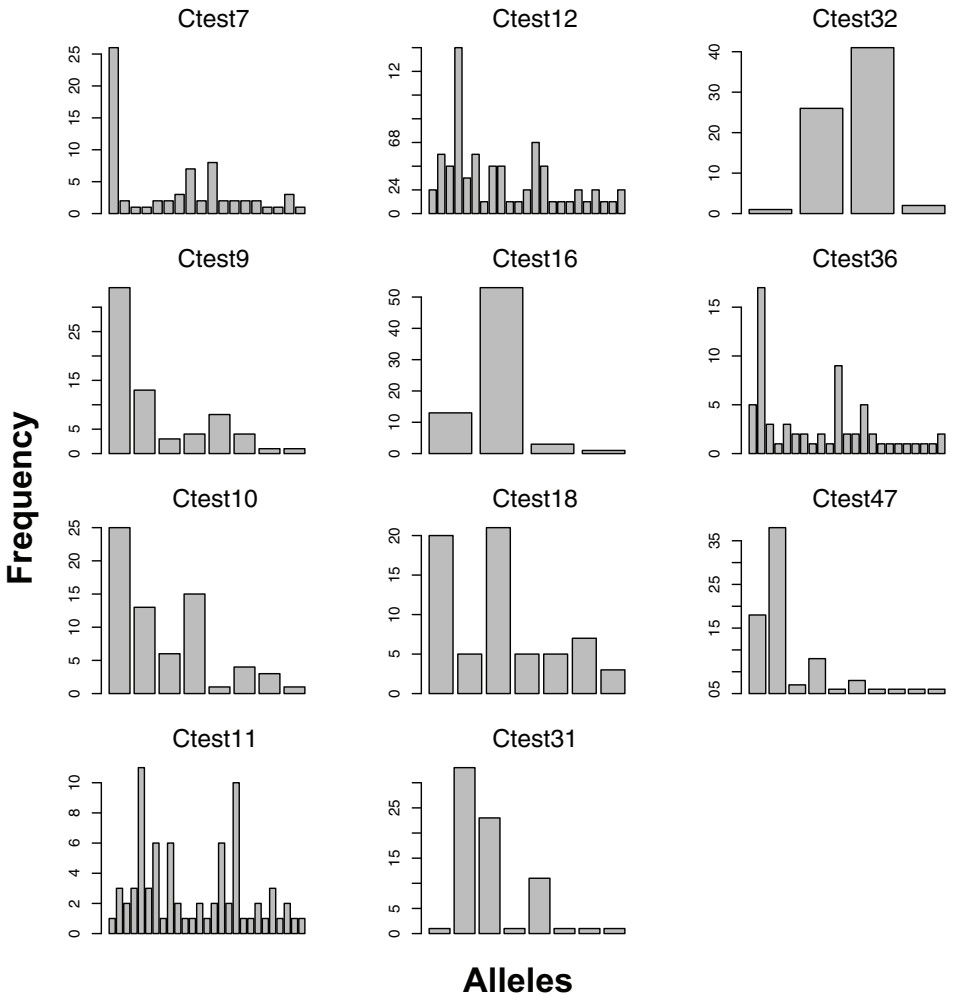

**Figure 1 Allele frequencies of each microsatellite locus amplified from the Atlantic lineage of *Chelonibia testudinaria*.** Each barplot represents a locus, each bar an allele, and the height of each bar indicates the frequency of each allele in in the data. Sample sizes are indicated in Table 2.

reproductive success is present, the effective population size of *C. testudinaria* should be low (*Hedgecock, 1994*). We estimated a population mutation rate (called theta, Θ) of 10 for the Atlantic *C. testudinaria* population using Watterson's estimator (*Watterson, 1975*) on published COI data, which suggests a large effective population size (data not shown). These data do not support the variance-in-reproductive-success hypothesis. The most likely cause for homozygote excess is the presence of null alleles. Null alleles are ubiquitous in microsatellite loci, and are caused by mutations in the primer sequence, leading to the dropout of alleles. While the true genotype is heterozygous, the observed genotype is homozygous due to the non-amplification of one of the alleles. Null alleles become increasingly prevalent with increasing effective population size (*Chapuis & Estoup, 2007*). *Chapuis & Estoup (2007)* show that simulated null allele frequencies were larger than 0.2 for all loci when Θ was one, the largest value they simulated. We estimated null allele frequencies between 0 and 0.3 for our microsatellite loci, well within the

**Table 2 Microsatellite loci characterization for *Chelonibia testudinaria* of the Atlantic lineage.** Range refers to the smallest (min) and largest (max) allele observed. Frequency of null alleles was estimated after *Brookfield (1996)*. Significance values of HWE test were adjusted for multiple comparisons (*Holm, 1979*). Genotyping error rates were based on re-genotyping of all Atlantic individuals.

| Locus | n | Range min | Range max | Number of alleles | Obs het. | Exp het. | HWE p-value | Allelic richness | Frequency null alleles | Genotyping error rate |
|---|---|---|---|---|---|---|---|---|---|---|
| Ctest7 | 34 | 206 | 314 | 18 | 0.56 | 0.82 | 0.09 | 16.11 | 0.14 | 0 |
| Ctest9 | 34 | 388 | 432 | 8 | 0.62 | 0.69 | **0.03** | 7.45 | 0.04 | 0.02 |
| Ctest10 | 34 | 264 | 278 | 8 | 0.65 | 0.77 | 0.13 | 7.45 | 0.07 | 0.03 |
| Ctest11 | 38 | 140 | 318 | 27 | 0.87 | 0.93 | 0.52 | 22.68 | 0.03 | 0.07 |
| Ctest12 | 34 | 388 | 476 | 23 | 0.74 | 0.92 | **0.01** | 20.26 | 0.1 | 0.02 |
| Ctest16 | 35 | 355 | 367 | 4 | 0.37 | 0.39 | 0.64 | 3.7 | 0.01 | 0 |
| Ctest18 | 33 | 450 | 485 | 7 | 0.61 | 0.78 | 0.18 | 6.98 | 0.1 | 0.06 |
| Ctest31 | 36 | 292 | 336 | 8 | 0.56 | 0.66 | **< 0.0001** | 6.54 | 0.07 | 0.07 |
| Ctest32 | 35 | 316 | 464 | 4 | 0.57 | 0.52 | 0.92 | 3.65 | 0 | 0 |
| Ctest36 | 33 | 336 | 524 | 23 | 0.46 | 0.89 | **< 0.0001** | 20.07 | 0.23 | 0.03 |
| Ctest47 | 37 | 255 | 385 | 10 | 0.62 | 0.66 | 0.78 | 8.36 | 0.02 | 0.02 |

**Note:**
n, number of individuals; Obs het., observed heterozygosity; Exp het., expected heterozygosity.

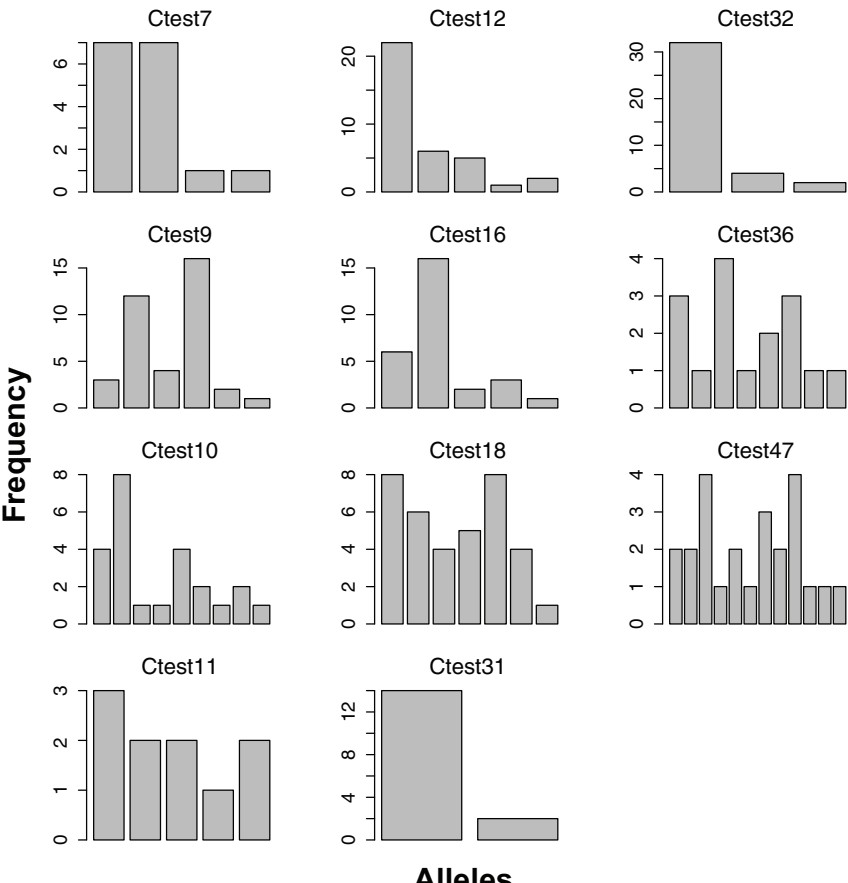

**Figure 2 Allele frequencies of each microsatellite locus amplified from the Indo-West Pacific lineage of *Chelonibia testudinaria*.** Each barplot represents a locus, each bar an allele, and the height of each bar indicates the frequency of each allele in in the data. Sample sizes are indicated in Table 3.

**Table 3 Microsatellite loci characterization for *Chelonibia testudinaria* of the Indo-West Pacific lineage.** Range refers to the smallest (min) and largest (max) allele observed. Frequency of null alleles was estimated after *Brookfield (1996)*. Significance values of HWE test were adjusted for multiple comparisons (*Holm, 1979*).

| Locus | n | Range min | Range max | Number of alleles | Obs het. | Exp het. | HWE p-value | Allelic richness | Frequency null alleles |
|---|---|---|---|---|---|---|---|---|---|
| Ctest7 | 8 | 186 | 314 | 4 | 1 | 0.61 | 0.13 | 3.07 | 0 |
| Ctest9 | 19 | 408 | 456 | 6 | 0.74 | 0.7 | 1 | 3.95 | 0 |
| Ctest10 | 12 | 260 | 344 | 9 | 0.5 | 0.81 | 0.08 | 5.48 | 0.17 |
| Ctest11 | 5 | 136 | 236 | 5 | 0.2 | 0.78 | 0.11 | 4.63 | 0.33 |
| Ctest12 | 18 | 354 | 374 | 5 | 0.33 | 0.58 | **< 0.0001** | 3.38 | 0.15 |
| Ctest16 | 14 | 351 | 375 | 5 | 0.57 | 0.61 | 1 | 3.53 | 0.02 |
| Ctest18 | 18 | 440 | 480 | 7 | 0.28 | 0.83 | **< 0.0001** | 5.22 | 0.3 |
| Ctest31 | 8 | 224 | 304 | 2 | 0.25 | 0.22 | 1 | 1.8 | 0 |
| Ctest32 | 19 | 308 | 316 | 3 | 0 | 0.28 | 0.08 | 2.13 | 0.22 |
| Ctest36 | 8 | 348 | 468 | 8 | 0.5 | 0.84 | 0.1 | 5.74 | 0.18 |
| Ctest47 | 12 | 140 | 340 | 12 | 0.83 | 0.89 | 0.14 | 6.88 | 0.03 |

**Note:**

n, number of individuals; Obs het., observed heterozygosity; Exp het., expected heterozygosity.

range of simulated data with large effective population size. Thus, the observed homozygote excess can be explained by the presence of null alleles.

We were able to amplify all loci in both the Atlantic and Indo-West Pacific lineage, which was somewhat surprising given large effective population size and significant between-lineage divergence. Both factors increase the chance for mutation accumulation in the primer sequences between lineages. Further, results for the Indo-West Pacific lineage need to be evaluated with caution because primers were designed from an individual of the Atlantic lineage. As expected by the concept of ascertainment bias (*Li & Kimmel, 2013*), the Atlantic lineage had higher allelic richness than the Indo-West Pacific lineage. However, fewer loci violated the expectations of HWE in the Indo-West Pacific lineage, likely an artifact of analyzing fewer individuals, resulting in low power to detect significant deviations from HWE. Future studies should increase sample sizes for both lineages to compare and contrast genotypic diversity.

Microsatellite markers have been developed successfully in a number of other barnacle species (*Dufresne, Parent & Bernatchez, 1999*; *Dufresne, Bourget & Bernatchez, 2002*; *Dawson et al., 2010*; *Plough & Marko, 2013*), providing tools to address interesting ecological and evolutionary questions. We expect the loci developed for *C. testudinaria* will be similarly useful.

### Funding

We received funding from the National Science Foundation (NSF-OCE No. 1029526) and the University of Georgia Department of Genetics Hightower Award. The funders had no role in study design, data collection and analysis, decision to publish, or preparation of the manuscript.

## Grant Disclosures

The following grant information was disclosed by the authors:
National Science Foundation: NSF-OCE No. 1029526.

## Competing Interests

The authors declare that they have no competing interests.

## Author Contributions

- Christine Ewers-Saucedo conceived and designed the experiments, performed the experiments, analyzed the data, wrote the paper, prepared figures and/or tables.
- John D. Zardus contributed reagents/materials/analysis tools, reviewed drafts of the paper, specimens.
- John P. Wares conceived and designed the experiments, contributed reagents/materials/analysis tools, reviewed drafts of the paper.

## Field Study Permissions

The following information was supplied relating to field study approvals (i.e., approving body and any reference numbers):
I sampled under the sampling permit of the University of Georgia Marine Institute.

## DNA Deposition

The following information was supplied regarding the deposition of DNA sequences:
NCBI short read archive:
STUDY: PRJNA310774
SAMPLE: SAP12-08-01
EXPERIMENT: Ctest_MiSeq (SRX1559760)
RUN: M00313 (SRR3144544).

## Data Deposition

We provided the data set as Supplementary Information.

## Supplemental Information

Supplemental information for this article can be found online at http://dx.doi.org/10.7717/peerj.2019#supplemental-information.

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
