# Peer review of "Microsatellite loci discovery from next-generation sequencing data and loci characterization in the epizoic barnacle Chelonibia testudinaria (Linnaeus, 1758)"

_PeerJ, doi:10.7717/peerj.2019_

## Round 0.1 · original submission · Minor Revisions

Please address the issues raised by the reviewers and provide me with your response in point-to-point format when you submit your revised version.

Reviewer 1 ·

Basic reporting

This manuscript describes the development and characterization of microsatellite markers for the epizoic barnacle Chelonibia testudinaria using shot-gun sequencing. Overall I found the paper to be well written and appropriate length (rather concise) given the nature of the subject (essentially a primer note). Presentation of the new primers and results of the initial population genetic work are largely sound, but I have some questions about details of the methods and results, which will need to be addressed before the MS can be accepted.

Experimental design

A couple of methods-related issues.

1) What software was used to map reads to Balanoides genome? What mis-match or identity cut-offs (or other input options) were used?

2) Were the usats that you chose to optimize perfect repeats or not? How were the 48 chosen? e.g. by motif or randomly?

Validity of the findings

Results:
I would like to see motif data (sequence and if compound, how many of each repeat motif) for each marker presented in a main text table or at least a supplementary table. It would just be another column in table 1 or added to the 'kmer' column.

Figure 1 needs more information I think. First, what is the Y-axis? I assume counts, but should be explicit in the figure. Note that one of the Y-axes has a decimal point - the others do not. Also, you should state in the figure that these data are from the Atlantic population. And that raises the question, why plot the Atlantic data and not the Indo-pacific data? Or why not plot them together?

You state that none of the loci had significant HWE deviations in the Indo-Pacific (line 154-155), but I see at least 2 loci that look to be significant in table 2 (Ctest31 and Ctest36) based on HWE P-Values of “0”. What are the digits on “0”? Is it so low that it is effectively zero, or is it a very small fraction close to zero? If it is VERY small, perhaps you should present as “<0.0001” – the “0” is hard to interpret especially since all other P values have decimal points.

Also, I see at least one case of significant heterozygote excess (Atlantic – Ctest7 ). This could be mentioned in results as currently only homozygote excess is described.

I am confused by Line 155 “richness was significantly lower than in the Atlantic individuals”. Allelic richness (AR) appears to be much HIGHER in the indo pacific (AR range ~ 3-22 in indo vs 1.8-6.88 Atlantic; Tables 2 and 3). Is this just a typo or am I missing something? You state that it is “significantly” higher …. Was this tested statistically?

Discussion:
Perhaps some discussion of the differences in diversity between the Atlantic and Indo-pacific populations would be good as diversity (allelic richness) appear to be very different. A sentence here could be useful. Given that usats were developed in the Atlantic lineage, you might expect a higher frequency of null alleles and thus reduced observed heterozygosity in the indo-pacific population, but we do not. Differences in diversity thus appear to be real, atleast for the sample sizes used.

Finally, there is little discussion about development of microsatellite markers using “next-gen” approaches in other barnacles, however there is at least one other paper with similar analyses and products (Plough and Marko 2014, Characterization of Microsatellite Loci and Repeat Density in the Gooseneck Barnacle, Pollicipes elegans, Using Next Generation Sequencing, J. Hered. 105). Perhaps there is no reason to refer to this paper in a comparative manner, but in the intro or discussion, some mention could be made as it uses similar techniques and presents similar results/analyses.

Reviewer 2 ·

Basic reporting

This manuscript is well written. The background is appropriate and relevant literature is properly referenced.

Experimental design

The epizoic barnacle Chelonibia testunaria is a non-model species that possess an unusual sexual system: androdiecy. Few genomic resources exist for this species, hence this study fill in this gap by providing 11 microsatellite loci that should prove useful for population genetic studies. The descriptive study has been well conducted and I have no major issue with it.

Validity of the findings

The sequencing and in silico analyses have been well conducted. Only 12 loci amplified and scored successfully out of 48, which is typical in this kind of study. Some loci showed homozygote excess in the Atlantic but not in the Pacific population. These microsatellite loci should prove useful in future studies.

Additional comments

This ms by Ewers-Saucedo et al. describes the discovery of thousands of microsatellite loci from illumina sequencing in the barnacle, Chelonibia testudinaria. Of these thousands loci, 11 were characterized and appear useful for population genetic studies. This approach is widely used and has proven successful for microsatellite discovery in a variety of species. As there is a paucity of genomic resources for this non-model species, this information will be useful for researchers wanting to assess the impact of this unusual sexual system.
I have a few comments for the authors:

L 133. Best not to comment on haploid genome size from sequencing data as it could be misleading.

L 153 Please provide number of individuals out of 23 that amplified successfully

Should we really expect HWE in epizoic animals ?

I find it a bit strange that null alleles were present in the Altantic individuals but not in the Pacific ones. Any ideas ?


Figure 1. i don't understand why the number of alleles don't correspond to table 2. For example, table 2 reports 18 alleles in Ctest7 but only 4 are reported on the graph.

Table 1. Please provide the repeat motifs for these microsatellite loci.
Were the allele sizes consistent with what was expected from the repeat number i.e. for tetranucleotides, did the alleles jump every 4 bp ?

Table 2. Did the authors use the bonferroni corrections to account for the 11 loci.

Did the authors test linkage disequilibrium among loci ?

L213 collèction should read collection

Hedgecock, and Jarne and Lagoda references. No parenthese should appear in the year of publication

---

## Round 0.2 · accepted · Accept

I am happy with the revised ms and pleased to inform you that your ms has been accepted for publication in PeerJ.

Thank you for submitting your ms to this journal.

Regards.